A quantitative approach for integrating multiple lines of evidence for the evaluation of environmental health risks

Schleier III Jerome J.
Marshall Lucy A.
Davis Ryan S.
Peterson Robert K.D. bpeterson@montana.edu
Department of Land Resources and Environmental Sciences, Montana State University , Bozeman, MT , USA
Hartung Thomas
Electronic publication date: 2015 Jan 15
Publication date: 2015
Volume: 3
Electronic Location ID: e730
Received 2014 Jul 23; Accepted 2014 Dec 29
Copyright: © 2015 Schleier III et al.
Copyright year: 2015
Copyright holder: Schleier III et al.
License: This is an open access article distributed under the terms of the Creative Commons Attribution License, which permits unrestricted use, distribution, reproduction and adaptation in any medium and for any purpose provided that it is properly attributed. For attribution, the original author(s), title, publication source (PeerJ) and either DOI or URL of the article must be cited.
License URL: https://creativecommons.org/licenses/by/4.0/

Keywords: Decision analysis, Uncertainty analysis, Mosquito management, Pesticide, Bayesian Markov Chain Monte Carlo, Risk assessment

Funding: USDA Western Regional IPM grant program 2009-34103-20034 Montana State University Institute on Ecosystems National Science Foundation U.S. Armed Forces Pest Management Board’s Deployed War Fighter Protection Research Program W911QY-11-1-0005 Montana Agricultural Experiment Station, Montana State University This research was supported by grants and fellowships from the USDA Western Regional IPM grant program (2009-34103-20034), Montana State University Institute on Ecosystems National Science Foundation Final Year Ph.D. Fellowship, the U.S. Armed Forces Pest Management Board’s Deployed War Fighter Protection Research Program (W911QY-11-1-0005), and by the Montana Agricultural Experiment Station, Montana State University, Bozeman, Montana, USA. The funders had no role in study design, data collection and analysis, decision to publish, or preparation of the manuscript.

==============================
Decision analysis often considers multiple lines of evidence during the decision making process. Researchers and government agencies have advocated for quantitative weight-of-evidence approaches in which multiple lines of evidence can be considered when estimating risk. Therefore, we utilized Bayesian Markov Chain Monte Carlo to integrate several human-health risk assessment, biomonitoring, and epidemiology studies that have been conducted for two common insecticides (malathion and permethrin) used for adult mosquito management to generate an overall estimate of risk quotient (RQ). The utility of the Bayesian inference for risk management is that the estimated risk represents a probability distribution from which the probability of exceeding a threshold can be estimated. The mean RQs after all studies were incorporated were 0.4386, with a variance of 0.0163 for malathion and 0.3281 with a variance of 0.0083 for permethrin. After taking into account all of the evidence available on the risks of ULV insecticides, the probability that malathion or permethrin would exceed a level of concern was less than 0.0001. Bayesian estimates can substantially improve decisions by allowing decision makers to estimate the probability that a risk will exceed a level of concern by considering seemingly disparate lines of evidence.

Introduction

Modeling and decision theory are being used increasingly for comparative and uncertainty analysis in risk management (Ascough et al., 2008). Researchers have advocated for a quantitative weight-of-evidence approach for estimating environmental risks from stressors such as contaminated sites and pesticides so that decision makers can comprehensively consider all evidence (Dale et al., 2008; Weed, 2005). The U.S. National Research Council (NRC) found that the U.S. Environmental Protection Agency (USEPA) needs to develop methods to address and communicate uncertainty and variability in all phases of the risk assessment process (National Research Council, 2009). The NRC stated that “Uncertainty forces decision makers to judge how probable it is that risks will be overestimated or underestimated for every member of the exposed population…” (National Research Council, 1994). In particular, the NRC reports found that, depending on the risk-management options, a quantitative treatment of uncertainty and variability is needed to discriminate between management options to make informed decisions (National Research Council, 1994; National Research Council, 1996; National Research Council, 2009).

When making decisions regarding risk, there are often multiple lines of evidence that need to be considered. Information often is generated and gathered from different sources, so risk analysts and managers are confronted with the issue of combining data from these sources to improve the decision-making process. However, the ability of people to make precise and significant statements about risks diminishes with increasing amounts of information and complexity (Zadeh, 1965). The incorporation of multiple lines of evidence into a weight-of-evidence framework allows risk assessors and managers to generate a single estimate of the risk (Dale et al., 2008). Currently, the most common way to incorporate dissimilar lines of evidence is by determining the weight-of-evidence estimate through qualitative risk assessments or through listing evidence (Chapman, McDonald & Lawrence, 2002; Hull & Swanson, 2006; Linkov et al., 2009; Menzie et al., 1996; Sanchez-Bayo, Baskaran & Kennedy, 2002; Suter II & Cormier, 2011; United States Environmental Protection Agency, 2005a; Weed, 2005), which can have fundamental mathematical limitations compared to quantitative estimates (Cox Jr, Babayev & Huber, 2005). These methods are important contributions to the decision making process, but they do not provide a comprehensive and structured approach for integrating multiple lines of evidence from different study types (Linkov et al., 2009).

Rather than testing for a specific relationship (e.g., the probability of obtaining values as extreme or more extreme than the values observed in the study), decision makers may ultimately be interested in making inferential conclusions about environmental health risks (Assmuth & Hilden, 2008; Ellison, 1996; Hill, 1996). Bayesian inference can address inferential conclusions by providing a framework, based on probability calculus, by quantifying the uncertainty in parameter estimates and determining the probability that an explicit endpoint is exceeded given a set of data (Ellison, 1996; Hill, 1996). Bayesian inference is a way of updating prior knowledge given new information becoming available to generate a posterior estimate of the parameters of interest (i.e., risk) (Ellison, 1996).

Currently there are few quantitative frameworks that integrate data into a framework that can be utilized by risk managers (Assmuth & Hilden, 2008). A quantitative framework for integrating and interpreting multiple lines of seemingly disparate evidence into an overall risk estimate is critically needed for complex risk assessments (Dale et al., 2008).

Risk assessment, biomonitoring, and epidemiology studies quantitatively estimate the likelihood that exposures to chemicals of interest exceed a threshold of observable effect or increased exposure over background levels in a population (McKone, Ryan & Ozkaynak, 2009). Epidemiological and biomonitoring data can play an important role in hazard identification and can also be considered in the risk characterization phase of the risk assessment process (Samet, Schnatter & Gibb, 1998). Therefore, the three seemingly disparate study methods are deriving an estimate of risk given exposure to the chemical of interest. Bayesian inference provides a quantitative framework for integrating these multiple lines of evidence into an overall estimate. Similar approaches have been used for different applications in risk assessment, toxicology, and environmental modeling, but they have not been utilized to update the risk estimates for anthropogenic chemical stressors as new information becomes available (Bernillon & Bois, 2000; Brand & Small, 1995; Devine & Qualters, 2008; Schenker et al., 2009; Taylor, Evans & McKone, 1993).

There are many advantages of using Bayesian techniques for weighing evidence, including full allowance for all parameter uncertainty in the model, the ability to include other pertinent information that would otherwise be excluded, and the ability to extend the models to accommodate more complex models (Hill, 1996; Sutton & Abrams, 2001). Studies utilizing Bayesian approaches have considered separate studies with the same study type to estimate an overall value for the parameter of interest (Smith, Lipkovich & Ye, 2002; Wheeler & Bailer, 2009). Therefore, to address the need for a quantitative approach for environmental health, we utilized Bayesian Markov Chain Monte Carlo (MCMC) to provide a logical and consistent method for estimating the risk of chemicals when multiple studies are available. To demonstrate how Bayesian statistics can be used for decisions regarding environmental and public health risks, we chose insecticides used for adult mosquito management as our case study.

Case Study

To effectively manage infection rates, morbidity, and mortality due to mosquito-borne pathogens, there must be a reduction in contact between infected mosquitoes and humans and animals (Marfin & Gubler, 2001). One of the more effective ways of managing high densities of adult mosquitoes that vector human and animal pathogens is ultra-low-volume (ULV) aerosol applications of insecticides. Since West Nile virus (WNV) was introduced into the U.S., more areas of the country have been experiencing large-scale insecticide applications. Consequently, there has been greater public attention on human-health and environmental risks associated with ULV insecticide applications (Peterson, Macedo & Davis, 2006; Reisen & Brault, 2007; Roche, 2002; Thier, 2001).

A decade after the initial response to WNV, several quantitative human-health and ecological risk assessments have been conducted to estimate the magnitude of risks associated with the insecticides (Davis, 2007; Davis, Peterson & Macedo, 2007; Gosselin et al., 2008; Macedo, Peterson & Davis, 2007; New York City Department of Health, 2005; Peterson, Macedo & Davis, 2006; Schleier III, 2008; Schleier III et al., 2009a; Schleier III et al., 2008a; Schleier III et al., 2009b; Schleier III et al., 2008b; Suffolk County, 2006; United States Environmental Protection Agency, 2005b; United States Environmental Protection Agency, 2005c; United States Environmental Protection Agency, 2005d; United States Environmental Protection Agency, 2006a; United States Environmental Protection Agency, 2006b; United States Environmental Protection Agency, 2006c; Valcke, Gosselin & Belleville, 2008). Also, there have been epidemiology and biomonitoring studies measuring the health effects after potential exposure to mosquito adulticides (Currier et al., 2005; Duprey et al., 2008; Karpati et al., 2004; Kutz & Strassman, 1977; O’Sullivan et al., 2005). Most studies suggest negligible public health risks from exposure to adulticides; however, no study has quantitatively combined the results from risk assessment, epidemiology, and biomonitoring studies, and their seemingly disparate data metrics, to obtain an overall estimate of the risk.

Data and Methods

In environmental and human health risk assessments of pesticides, risk quotients (RQ) are often used to quantitatively express risk (Peterson, 2006). Risk quotients are calculated by dividing the potential exposure (PE) by its respective toxic endpoint value. Estimated RQs are compared to a RQ level of concern (LOC) or other threshold which is set by the USEPA or another regulatory agency to determine if regulatory action is needed. The RQ LOC used in our assessment was 1.0. An RQ > 1.0 means that the estimated exposure is greater than the relevant toxicological endpoint. If an RQ breaches a regulatory LOC (RQ ≥ 1) at a lower tier, then risk managers decide to restrict the product use, progress to higher tier risk assessments, or use field-verified models (United States Environmental Protection Agency, 2006d).

We chose two pesticides for our case study, malathion (O,O-dimethyl dithiophosphate of diethyl mercaptosuccinate) and permethrin ([3-phenoxyphenyl]methyl 3-[2,2-dichloroethenyl]-2,2-dimethylcyclopropane carboxylate), because biomonitoring, epidemiology, and risk assessments have been performed with respect to ULV applications for adult mosquito management (Table 1). We chose adult human males for our case study because it is the only common group assessed by all studies. To ensure that we possessed all publically available studies, a literature review was performed and all relevant studies were pulled from government reports and academic journals from 1900 to 2014 using the Google and Thomas Reuters Web of Science™ search engines. All studies that we found that contained mosquito ULV risk assessments, biomonitoring, or epidemiological measurements for permethrin or malathion were included in this assessment.

Table 1 Risk quotient estimates for each study.

	Malathion	Permethrin	
Karpati et al. (2004) c	NAa	0.99b	
United States Environmental Protection Agency (2005c) d and
United States Environmental Protection Agency (2005d) d	0.018	0.025	
Currier et al. (2005) e	NAa	0.99b	
O’Sullivan et al. (2005) c	0.99b	NAa	
Peterson, Macedo & Davis (2006) d	0.0076	0.0021	
Suffolk County (2006) d	0.015	0.013	
Macedo, Peterson & Davis (2007) d	NAa	0.023	
Valcke, Gosselin & Belleville (2008) d	0.64	NAa	
Schleier III (2008) d	NAa	0.00025	
Schleier III et al. (2009a) d	0.02	NAa	
Schleier III et al. (2009b) d	0.0017	0.000068	
Notes.

a Not applicable because the chemical was not assessed.

b A risk quotient of 0.99 was used because it provides a conservative estimate of the risk for biomonitoring and epidemiology studies and due to a lack of knowledge about the true value, which must be below 1 if no effect is seen.

c Epidemiological study.

d Risk assessment.

e Biomonitoring study.

The estimated RQs for each study are summarized in Table 1 for each chemical. The same toxicological endpoints were used for all of the risk assessments, which are based on the U.S. EPA’s ingestion reference dose for mammals (Macedo, Peterson & Davis, 2007; Peterson, Macedo & Davis, 2006; Schleier III, 2008; Schleier III et al., 2009a; Schleier III et al., 2009b; Valcke, Gosselin & Belleville, 2008), and in the case of probabilistic risk assessments we used the 95th percentile RQ for conservatism.

The literature search found two epidemiological studies and one biomonitoring study for permethrin and malathion. Karpati et al. (2004) analyzed hospital admissions for asthma in New York, NY three days before and after ground based ULV applications of permethrin (n = 510 before spraying and 501 after spraying) and found no increase in admissions for asthma. Currier et al. (2005) analyzed urine samples for metabolites of permethrin in 125 persons in the treated area and 67 persons from two control areas after ground-based ULV applications in Mississippi and found no increase in urinary metabolites. The persons selected in the study were geographically random and were verified by mapping the GPS location of the ground-based applications. O’Sullivan et al. (2005) analyzed hospital admissions for asthma in New York, NY after ground-based ULV applications of malathion in September of 1999, and compared those to September 1997 and 1998 when no malathion treatments occurred (n = 1,318 patients presented with a diagnosis of asthma exacerbation). They found no statistical difference between the 1999 asthma admissions and the asthma admission in 1997 and 1998. To incorporate the epidemiology and biomonitoring studies, we assumed that if the researchers did not observe an effect or increase in urinary metabolites of the pesticide, the RQ was assumed to be 0.99 (Table 1). We assumed a RQ of 0.99 to be conservative because of a lack of knowledge on the value, which must be below 1.0 if no effect is observed.

Bayesian inference treats statistical parameters as random variables, and uses a likelihood function to express the plausibility of obtaining different values of the parameter when the data have been observed (Ellison, 1996). To define a RQ for adult males we used Bayes’ theorem: (1) pθ|y=py|θpθ

where p is the probability mass, θ is the value of a random variable selected from the prior distribution, y is the evidence being considered, p(θ) is the prior probability, p(y|θ) is the likelihood function for the evidence (Congdon, 2006; Gelman et al., 2004). We assumed a normal distribution for the likelihood function and used log-transformed risk quotients from Table 1. The central limit theorem of classical statistics and the Bayesian analog justify the normal density as an approximation for the posterior distribution of many summary statistics, even when they are derived from non-normal data (Congdon, 2006). To estimate the posterior density, (2) py|θ=12πσexp−12σ2y−θ2

where y is a single scalar observation from the RQ’s in Table 1 from a normal distribution parameterized by a mean of θ and a variance of σ2 (Gelman et al., 2004).

We have no knowledge of the prior distribution, so we assumed an uninformative or diffuse prior which we defined as a normal distribution with a μ0 of 1 and a τ02 of 1. We chose an uninformative prior because the effect of the prior and data on the updated beliefs depends on the precision of the density of p(θ) (Congdon, 2008). We used MCMC simulation utilizing the Metropolis-Hasting algorithm to obtain the posterior distribution for Eq. (2) using Matlab® R2010b (MathWorks, Natick, MA, USA). We sampled the purposed posterior distributions using Eq. (2) by iterating 100,000 purposed values for the posterior distribution and discarded the first 1,000 samples for burn in.

Results and Discussion

The mean posterior RQs after all studies were incorporated were 0.4386 with a variance of 0.0163 for malathion and 0.3281 with a variance of 0.0083 for permethrin (Figs. 1 and 2). The mean posterior RQs for all studies excluding the epidemiological and biomonitoring studies slightly decreased the mean to 0.4119 with a variance of 0.0158 for malathion and a mean of 0.302 with a variance of 0.0081 for permethrin (Figs. 1 and 2). Using the posterior mean and variance from the normal distribution, the probability that exposure to malathion or permethrin after ULV applications would exceed a level of concern was less than 0.0001, regardless of whether all of the studies were incorporated or the epidemiological and biomonitoring studies were held out (Figs. 1 and 2).

Figure 1 Posterior probability distributions for malathion with all available studies and all studies excluding epidemiological and biomonitoring.

Figure 2 Posterior probability distributions for permethrin with all available studies and all studies excluding epidemiological and biomonitoring.

The risk assessments used different data and exposure scenarios to estimate the RQ. The utility of the Bayesian inference for risk management is that the estimated RQ represents a probability distribution from which we can obtain a probability of exceeding a threshold (Figs. 1 and 2). The probability of exceeding a threshold is most likely more intuitive for risk managers and the public to understand than an estimate of the 95th percentile of exposure or risk, which is typically reported in probabilistic risk assessments (Hill, 1996). In fact, risk can be defined as the probability and severity of adverse effects (Aven & Renn, 2009), which Bayesian statistics directly addresses. The majority of weight-of-evidence studies do not quantify both a risk estimate and variability or uncertainty around that estimate, but Bayesian MCMC methods quantify both (Linkov et al., 2009).

The USEPA provides guidance on how to perform risk assessments that address variability and uncertainty (National Research Council, 2009; United States Environmental Protection Agency, 1989; United States Environmental Protection Agency, 2004), but they do not provide a simple method for integrating multiple lines of evidence. Our case study directly addresses the need for a standard approach by which multiple lines of evidence can be interpreted in a framework that ecologists, risk assessors and managers, and NRC have highlighted (Dale et al., 2008; Linkov et al., 2009; National Research Council, 1994; National Research Council, 1996; National Research Council, 2009). Our method also could be utilized by the Network Reference Laboratories for Monitoring of Emerging Environmental Pollutants in the European Union for integrating risk assessments and biomonitoring to prioritize pollutants (Tilghman et al., 2009).

The USEPA and other regulatory agencies potentially could benefit from using a value-of-information approach that takes advantage of Bayesian inference to determine if generating new data will significantly improve the risk estimate, similar to approaches used for toxicological studies (National Research Council, 2009; Taylor, Evans & McKone, 1993). Our analysis showed that the addition of epidemiological and biomonitoring studies using conservative estimates did not drastically change the estimate of risk. Biomonitoring assessments could provide a refined RQ estimate if the amount of chemical the person is exposed to is calculated. Bayesian inference can also incorporate expert knowledge of a system which can be used as prior information that is updated by data (Gargoum, 2001; Morris, 1977).

In ecotoxicology and other disciplines, there are multiple estimates of values like the lethal concentration that kills 50% of a population (LC50) (Wheeler & Bailer, 2009). This technique could be used to estimate an overall LC50 for use in risk assessments or setting total maximum daily load limits. Stauffer (2008) showed that in natural resource management there are often multiple estimations for a population of interest. Therefore, Bayesian MCMC methods can be used to estimate the probability of the population being above or below a given threshold.

Bayesian analysis provides a systematic approach for guiding the decision-making process by incorporating new knowledge in the estimate of risk, which directly addresses NRC recommendations (National Research Council, 1994; National Research Council, 2009). However, Bayesian inference does not address the uncertainties inherent in each risk assessment. For example, there is large uncertainty surrounding the estimate of insecticide air concentrations and deposition on surfaces after ULV applications for adult mosquito management (Schleier III et al., 2009a; Schleier III et al., 2009b). Models used by the USEPA and other researchers to estimate concentrations are either over- or under-estimating depending on the model (Schleier III & Peterson, 2010; Schleier III et al., 2008b). In addition, probabilistic risk assessments demonstrated that the estimated air concentration and deposition of insecticides are contributing the largest amount of variance to the potential exposure (Schleier III et al., 2009a; Schleier III et al., 2009b). However, the estimate presented here most likely is robust against these uncertainties because the studies used a variety of models, exposure pathways, and monitoring techniques which were not dependent on a standardized assessment protocol.

We recognize that the assumptions about RQ distributions may affect the final results; however, we attempted to reduce the potential biases by making conservative assumptions erring on the side of safety, which is common practice in risk assessment. In addition, probability distributions other than normal can be utilized if enough is known about the underlying distribution of the population, like those used for toxicological studies. Bayesian MCMC also can be utilized with the current data and the incorporation of expert judgments to aid in the determination of risk estimates (Grist et al., 2005).

Bayesian analysis techniques have been underutilized with respect to environmental and public health, risk assessment, ecology, and environmental sciences (Clark, 2005). Our method is a quantitative approach to statistically derive risk estimates from multiple lines of evidence, which is a relatively simple way of integrating multiple lines of evidence into a framework that can be used by assessors and managers (Assmuth & Hilden, 2008; Linkov et al., 2009). In addition to insecticide risk, this approach can be used for other anthropogenic agents such as dioxins and polychlorinated biphenyls, which in many cases have risk assessment, biomonitoring, and epidemiology studies performed for a site. The method presented here can also be utilized for probabilistic ecological risk assessments to derive a distribution for the toxicological endpoints like LC50 or no-effect concentration when multiple values are available for the same species. Future refinements to our Bayesian model would be the development of a method to convert epidemiological study results into a RQ to reduce the uncertainty and conservatism. In addition, biomonitoring studies can quantify the exposure (if exposures are above background levels) and convert those estimates to RQ.

We thank K Irvine (U.S. Geological Survey) for reviewing an earlier version of the manuscript.

Additional Information and Declarations

Competing Interests

Author Contributions

Ryan S. Davis is an employee of Electrical Consultants, Inc.

Jerome J. Schleier III conceived and designed the experiments, performed the experiments, analyzed the data, contributed reagents/materials/analysis tools, wrote the paper, prepared figures and/or tables, reviewed drafts of the paper.

Lucy A. Marshall analyzed the data, contributed reagents/materials/analysis tools, wrote the paper, reviewed drafts of the paper.

Ryan S. Davis conceived and designed the experiments, contributed reagents/materials/analysis tools, reviewed drafts of the paper.

Robert K.D. Peterson conceived and designed the experiments, wrote the paper, reviewed drafts of the paper.

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

United States Environmental Protection Agency (1989) United States Environmental Protection Agency Risk assessment guidance for superfund. Volume I. Human health evaluation manual (Part A) 1989 Washington D.C. Environmental Protection Agency
United States Environmental Protection Agency (2004) United States Environmental Protection Agency Risk assessment principles and practices: staff paper 2004 Washington D.C. Environmental Protection Agency
United States Environmental Protection Agency (2005a) United States Environmental Protection Agency Guidelines for carcinogen risk assessment 2005a Washington D.C. Environmental Protection Agency
United States Environmental Protection Agency (2005b) United States Environmental Protection Agency Memorandum from B. Davis, Health Effects Division, to C. Rodia, Special Review and Registration. Re: occupational and residential exposure assessment and recommendations for the reregistration elegibility decision (RED) for piperonyl butoxide 2005b Washington D.C. Environmental Protection Agency
United States Environmental Protection Agency (2005c) United States Environmental Protection Agency Memorandum from S.L. Kinard, Health Effects Division, to T. Moriarty, Special Review and Reregistration Division. Malathion: updated revised human health risk assessment for the reregistration eligibility decision document (RED) 2005c Washington D.C. Environmental Protection Agency
United States Environmental Protection Agency (2005d) United States Environmental Protection Agency Memorandum from S.L. Kinard, Y. Yang and S. Ary, Health Effects Division, to J. Guerry, Special Review and Reregistration Division. Re: Permethrin. HED Chapter of the Reregistration Eligibility Decision Document (RED) 2005d Washington D.C. Environmental Protection Agency
United States Environmental Protection Agency (2006a) United States Environmental Protection Agency Interim reregistration eligibility decision for naled, Case No. 0092 2006a Washington D.C. Environmental Protection Agency 1 130
United States Environmental Protection Agency (2006b) United States Environmental Protection Agency Reregistration eligibility decision (RED) for permethrin 2006b Washington D.C. Environmental Protection Agency 1 95
United States Environmental Protection Agency (2006c) United States Environmental Protection Agency Reregistration eligibility decision for malathion 2006c Washington D.C. Environmental Protection Agency 1 101
United States Environmental Protection Agency (2006d) United States Environmental Protection Agency 2006d Technical overview of ecological risk assessment. Available at http://www.epa.gov/oppefed1/ecorisk_ders/toera_risk.htm (accessed 1 November 2006)
Valcke, Gosselin & Belleville (2008) Valcke M Gosselin N Belleville D Human exposure to malathion during a possible vector-control intervention against West Nile Virus. II: evaluation of the toxicological risks using a probabilistic approach Human and Ecological Risk Assessment 2008 14 1138 1158 10.1080/10807030802493891
Weed (2005) Weed DL Weight of evidence: a review of concept and methods Risk Analysis 2005 25 1545 1557 10.1111/j.1539-6924.2005.00699.x 16506981
Wheeler & Bailer (2009) Wheeler MW Bailer AJ Benchmark dose estimation incorporating multiple data sources Risk Analysis 2009 29 249 256 10.1111/j.1539-6924.2008.01144.x 19000080
Zadeh (1965) Zadeh L Fuzzy sets Information and Control 1965 8 338 353 10.1016/S0019-9958(65)90241-X