# Peer review of "A quantitative approach for integrating multiple lines of evidence for the evaluation of environmental health risks"

_PeerJ, doi:10.7717/peerj.730_

## Round 0.1 · original submission · Major Revisions

This is an interesting article, which, however, needs more details along the comments of the reviewers. There are a lot of important papers in the field not referred to and this draws on the conclusions.

Reviewer 1 ·

Basic reporting

The article is well and comprehensively written. Adding a Conclusion would be an option to improve the structure of the manuscript. It should be easily extracted from the Results and Discussions.

Experimental design

The manuscript aims at demonstrating the usefulness of Bayesian approaches for review of data relevant for human health risk management. As and example, two common insecticides are used. It concludes that Bayesian probability estimates can improve decision making substantially. It claims novelty in applying well-known analytical procedures to existing data. It is assumed that such a combination is within the scope of the journal as a) the academic editor considered the manuscript for review and b) the reviewer considers it as relevant research.
The authors should, however, add details on how the reviewed studies were retrieved. If not done systematically, they should considered to discuss the value of a systematic literature search. The studies used should be described in more detail (Table 1: Are 95%-quantiles reported? Do the a and b (in the legend) apply to all NA and 0.99? Add study type (risk assessment; biomonitoring, epidemiological study)!).
The option of a sensitivity analysis instead of conservatively choosing a default of 0.99 for cases without a value should be at least discussed.
Double-check the use of 'P()' and 'p()' !

Validity of the findings

No comment, except that 'Using the posterior mean and variance from the normal distribution...' in Results and Discussion is not clear and should be re-phrased.

Additional comments

The approach should be considered in the wider-frame of the use and usefulness of evidence-based approaches in risk management, see e.g. comment made on systematically revising literature. A general move towards these type of approaches is taking place, see e.g. toxicology.

Reviewer 2 ·

Basic reporting

Manuscript is clearly understandable, materials and methods are well described. Language, grammar etc. is fine.

No comments

Experimental design

For me it looks like that the authors used the total risk quotients of each study to build the Bayesian model without taking into account the sample sizes and variance estimates of the individual studies which can not result in a unbiased and best estimator.
The authors have not used any other study parameters to build there model which is a missed chance to improve the model.
The authors have not compared the different variables of the datasets: Are they studies sufficiently comparable to be combined? This would be a necessary approach for evidence-based medicine meta analysis.

Validity of the findings

see comments before

Additional comments

The introduction is to critical with regard to the current use of Bayesian approaches in risk and hazard assessment especially with regard to meta analysis like in this manuscript. There is a variety of publications in the last years where - especially with small sample sizes - Bayesian methods have been successfully used. The authors should modify the introduction that it reflect the latest developments and give some examples of applications.

Reviewer 3 ·

Basic reporting

Fine.

Experimental design

Fine.

Validity of the findings

Not clear (see below).

Additional comments

“A Quantitative Approach for Integrating Multiple Lines of Evidence for the Evaluation of Environmental Health Risks “by Schleier et al. describes a method that could prove to be very useful for risk assessors. On the one hand, this paper was very focused on the ultimate method, which makes it easier to follow. On the other hand, there are many things that the authors seem to imply should be considered fact that actually requires much more explanation.

One major concern is the risk quotients in Table 1. I find it hard to believe there is no subjectivity or uncertainty in these calculations. For example, is an epidemiology study assumed to have no effect if results are not statistically significant? How are confidence intervals considered? How do we know the most appropriate toxicity value is used? Is there controversy regarding interpretation of study results? This merits some discussion.

Also, how are all lines of evidence actually incorporated? I didn’t understand how they fit into the equation (which may be a result of my limited knowledge of Bayesian statistics, but I assume other readers will be in the same boat).

---

## Round 0.2 · Minor Revisions

I agree with reviewer 1 that the search strategy for the studies included should be described.

Reviewer 1 ·

Basic reporting

All standards met.

Experimental design

All earlier comments for addressed, with the exception of the suggestion to describe the literature search in more detail. This would be informative as this would allow the reader to judge potential biases.

Validity of the findings

Fine.

Additional comments

The suggestion to consider the approach in view of a broader development has not been taken up in the manuscript. It may beyond the authors' intention here, but should be kept in mind for future work.

Reviewer 2 ·

Basic reporting

No Comments

Experimental design

No Comments

Validity of the findings

Response Point 1:

I do not feel that my concerns have been at all addressed by the answers of the authors. The different studies have unbalanced sample sizes as now shown for a few of the studies in the manuscript, however, in my believe only the risk quotient estimates from table 1 that summarizes the studies have entered the model without taking into account uncertainty or sample size estimates of the individual studies as (weighting?) factors. As frequentist I am not really an expert in Bayesian approaches but I still feel that any meta analysis estimator that is not taking the individual study sample sizes and uncertainty/variability estimates into account can not be the correct approach in the sense of being the best estimator (lowest variance, unbiased). If the authors feel that my understanding of this is not correct I would like to urge the authors to explain this in the manuscript.
In addition I would like to ask the authors to calculate with their approach the mean posterior RQs of the following hypothetic study to illustrate the impact of sample size in their approach and share it with the editor and reviewers:

Study Risk Quotient Sample Size
a 0.018 500
b 0.015 1000
c 0.99 10


Response Point 2:
With the additional study summaries in data and methods and since the main purpose of the manuscript is to illustrate a quantitative approach for integrating multiple lines of evidence rather than a risk assessment of malathion and permethrin I find the answers of the authors satisfactory. However, in a real risk assessment publication I would insist that it is necessary for an evidence-based approach to define inclusion and exclusion criteria and by that define whether the studies are sufficiently comparable to be combined.

Additional comments

Response Point 3:
OK. No further comments.

Reviewer 3 ·

Basic reporting

No comments.

Experimental design

No comments.

Validity of the findings

No comments.

Additional comments

No comments.

---

## Round 0.3 · accepted · Accept

Thank for (re-)revising your manuscript, which is now acceptable for publication.
Thank you very much for publishing your interesting work in PeerJ.